# Ewing Sarcoma-Derived Extracellular Vesicles Impair Dendritic Cell Maturation and Function

**DOI:** 10.3390/cells10082081

**Published:** 2021-08-13

**Authors:** Hendrik Gassmann, Kira Schneider, Valentina Evdokimova, Peter Ruzanov, Sebastian J. Schober, Busheng Xue, Kristina von Heyking, Melanie Thiede, Guenther H. S. Richter, Michael W. Pfaffl, Elfriede Noessner, Lincoln D. Stein, Poul H. Sorensen, Stefan E. G. Burdach, Uwe Thiel

**Affiliations:** 1Department of Pediatrics, Children’s Cancer Research Center, Kinderklinik München Schwabing, School of Medicine, Technical University of Munich, 80804 Munich, Germany; hendrik.gassmann@tum.de (H.G.); kira.schneider@tum.de (K.S.); s.schober@tum.de (S.J.S.); busheng.xue@tum.de (B.X.); kristina.heyking@tum.de (K.v.H.); melanie.thiede@tum.de (M.T.); 2Ontario Institute for Cancer Research, Toronto, ON M5G 0A3, Canada; v.evdokimova@gmail.com (V.E.); pruzanov@gmail.com (P.R.); lincoln.stein@gmail.com (L.D.S.); 3Department of Molecular Oncology, British Columbia Cancer Research Centre, Vancouver, BC V5Z 1L3, Canada; psor@mail.ubc.ca; 4Division of Oncology and Hematology, Department of Pediatrics, Charité—Universitätsmedizin Berlin, 13353 Berlin, Germany; guenther.richter@charite.de; 5Division of Animal Physiology and Immunology, School of Life Sciences Weihenstephan, Technical University of Munich, 85354 Freising, Germany; michael.pfaffl@wzw.tum.de; 6Immunoanalytics Research Group—Tissue Control of Immunocytes, Helmholtz Center, 81377 Munich, Germany; noessner@helmholtz-muenchen.de; 7Department of Pathology and Laboratory Medicine, University of British Columbia, Vancouver, BC V5Z 1L3, Canada; 8Translational Pediatric Cancer Research, Institute of Pathology, Technical University of Munich, 80804 Munich, Germany; 9German Cancer Consortium (DKTK), German Cancer Research Center (DKFZ), Partner Site Munich, 81675 Munich, Germany

**Keywords:** myeloid cells, dendritic cells, Ewing sarcoma, extracellular vesicles, inflammation, immunosuppression, tumor microenvironment

## Abstract

Ewing sarcoma (EwS) is an aggressive pediatric cancer of bone and soft tissues characterized by scant T cell infiltration and predominance of immunosuppressive myeloid cells. Given the important roles of extracellular vesicles (EVs) in cancer-host crosstalk, we hypothesized that EVs secreted by EwS tumors target myeloid cells and promote immunosuppressive phenotypes. Here, EVs were purified from EwS and fibroblast cell lines and exhibited characteristics of small EVs, including size (100–170 nm) and exosome markers CD63, CD81, and TSG101. Treatment of healthy donor-derived CD33^+^ and CD14^+^ myeloid cells with EwS EVs but not with fibroblast EVs induced pro-inflammatory cytokine release, including IL-6, IL-8, and TNF. Furthermore, EwS EVs impaired differentiation of these cells towards monocytic-derived dendritic cells (moDCs), as evidenced by reduced expression of co-stimulatory molecules CD80, CD86 and HLA-DR. Whole transcriptome analysis revealed activation of gene expression programs associated with immunosuppressive phenotypes and pro-inflammatory responses. Functionally, moDCs differentiated in the presence of EwS EVs inhibited CD4^+^ and CD8^+^ T cell proliferation as well as IFNγ release, while inducing secretion of IL-10 and IL-6. Therefore, EwS EVs may promote a local and systemic pro-inflammatory environment and weaken adaptive immunity by impairing the differentiation and function of antigen-presenting cells.

## 1. Introduction

Ewing sarcoma (EwS) is the second most frequent bone and soft tissue tumor in children and young adults with high metastatic capacity. In the case of multifocal disease or early relapse, 5-year survival rates drastically decline to less than 30% [1,2]. EwS is a non-immunogenic tumor with a low mutational burden [3] and reduced expression of human leukocyte antigen (HLA) class I molecules [4]. Immunohistochemistry and transcriptome-based analysis revealed that the EwS tumor microenvironment (TME) is scarcely infiltrated with cytotoxic T cells [5,6], mature dendritic cells (DCs), and pro-inflammatory M1 macrophages [7,8,9]. Instead, EwS tumors are mostly populated by immature (M0) and immunosuppressive (M2) macrophages [7,8,9], whose abundance in the TME is correlated with worse outcomes [10]. However, mechanisms promoting the immunosuppressive TME in EwS are poorly understood.

One of the potential mechanisms may involve tumor-derived extracellular vesicles (EVs) which emerged in the past decade as mediators of tumor-associated inflammation and immunosuppression [11,12]. Small EVs range from 40–160 nm in size and include exosomes (originating from multivesicular endosome compartments) and heterogeneous populations of plasma membrane-derived microvesicles [13]. EVs are secreted by healthy tissue cells as well as by tumor cells, including EwS [14,15,16]. The EV cargo is protected by a lipid membrane and reflects the physiological or pathological state of the parental cell. It is mostly comprised of proteins, lipids, and RNA, which can be transferred to the recipient cell [13,17,18].

Circulating myeloid cells including monocytes can encounter and engulf tumor-derived EVs in peripheral blood or after extravasation into the tumor stroma. Monocytes can differentiate into DCs specializing in antigen processing and priming of CD4^+^ and CD8^+^ T cells, thereby activating adaptive immune responses including antitumor immunity [19]. However, proper differentiation and maturation of DCs can be disrupted by inflammatory signals [20]. Immature DCs are incapable of priming cytotoxic T cell responses, and, instead, promote T cell anergy, inflammation, and tumor progression [19,21]. Cancer-associated chronic inflammation can also lead to pathological activation of myeloid cells and expansion of myeloid-derived suppressor cell (MDSC) populations in the blood of cancer patients, including EwS [22,23]. MDSCs possess potent immunosuppressive functions and are involved in cancer, autoimmune, and chronic inflammatory diseases [24]. Tumor-derived EVs may play active roles in both chronic inflammation and conversion of myeloid cells into MDSCs and other immunosuppressive phenotypes [25]. The examples include polarization of myeloid cells into pro-tumorigenic M2 macrophages by glioblastoma-derived EVs [26] and induction of the immune checkpoint molecule PD-L1 on circulating monocytes by EVs from melanoma and chronic lymphatic leukemia [27,28].

Previously, we identified and characterized EVs isolated from EwS cell lines and patients’ plasma [14,29], indicating that similar mechanisms may be operating in EwS. We also established protocols for the generation of monocytic-derived DCs (moDCs) used in the priming of allo-restricted tumor antigen-specific T cells for adoptive T cell transfer [30,31,32]. In this study, we investigated if and how EwS-derived EVs may affect the phenotype and function of CD33^+^ and CD14^+^ myeloid cells isolated from the blood of healthy donors. We show that EwS EVs induce pro-inflammatory cytokine release and impair myeloid cell differentiation as well as their T cell stimulatory function, providing novel insights into the mechanisms of EwS-associated inflammation and immunosuppression.

## 2. Materials and Methods

### 2.1. EV-Isolation and Characterization

A4573, A673, TC32, and TC71 EwS cell lines and MRC5 fibroblast cell line (DSMZ; Braunschweig, Germany) were cultured in RPMI 1640 or DMEM/F12 medium supplemented with 10% fetal bovine serum (FBS), 100 U/mL penicillin and 100 µg/mL streptomycin (all obtained from Life Technologies, Grand Island, NY, USA). Cells were regularly tested to be free of mycoplasma contamination. For EV isolation, cells were seeded in ten 150 mm tissue culture dishes (Sarstedt, Newton, NC, USA). At 70–80% confluency, cells were washed with PBS and cultured for 18 h in a fresh medium containing 2% exosome-depleted FBS (Thermo Fisher Scientific, Vilnius, Lithuania). Conditioned medium was collected from the cells showing >90% viability, as confirmed by trypan blue staining. EVs were isolated by ultrafiltration and differential centrifugation as described previously [14,29]. Briefly, conditioned medium was sequentially centrifuged at 2000× g for 10 min, 10,000× *g* for 30 min, concentrated at 4200× *g* for 7 min using Amicon Ultra-15mL-30K tubes (Millipore, Merck Chemicals GmbH, Darmstadt, Germany), and passed through 0.22 µm filters (TPP, Trasadingen, Switzerland). Filtered conditioned medium was diluted 1:1 with ice-cold PBS and subjected to ultracentrifugation at 100,000× *g* for 120 min, followed by second centrifugation at 100,000× *g* for 120 min (S110-AT rotor, Thermo Scientific, k factor 78.4). Cleared EV pellets were resuspended in 200 µL ice-cold PBS. Each purification step was performed at 4 °C. For plasma EV isolation, blood from healthy donors collected into EDTA tubes was centrifuged at 10,000× *g* for 10 min at 4 °C to remove cellular components. Plasma supernatant was diluted 1:1 with ice-cold PBS, passed through 0.22 µm filters, and subjected to ultracentrifugation as described above. EVs were quantified by nanoparticle tracking analysis (NTA, ZetaView^®^, Particle Metrix, Inning am Ammersee, Germany) and further characterized by immunoblotting using antibodies against β-actin (sc-47778), calnexin (sc-23954), CD63 (sc-5275), CD81 (sc-7637), Hsp70 (sc-32239), Hsp90 (sc-69703), TSG101 (sc-7964), and β-tubulin (sc-5274; all antibodies obtained from Santa Cruz Biotechnology, Dallas, TX, USA).

### 2.2. Generation of moDCs

Peripheral blood mononuclear cells (PBMCs) were isolated from healthy donors (DRK-Blutspendedienst Baden-Wuerttemberg-Hessen, Ulm, Germany) using density-gradient separation (Ficoll-Paque, GE Healthcare, Freiburg, Germany). CD14^+^ or CD33^+^ myeloid cells were isolated by positive selection using anti-human CD14 magnetic particles (BD Biosciences, Heidelberg, Germany) or anti-human CD33 MicroBeads (Miltenyi Biotec, Bergisch Gladbach, Germany) according to the manufacturer’s protocol. The CD14-depleted fraction was stored at −80 °C until further use, see below. moDCs were generated as reported previously [30]. Briefly, CD14^+^ or CD33^+^ myeloid cells were differentiated in X-Vivo15 medium containing 1% human AB serum (both obtained from Lonza, Basel, Switzerland), 800 U/mL of recombinant human (rh) GM-CSF (Sanofi, Bridgewater, NJ, USA), and 30 ng/mL rhIL-4 (R&D Systems, Wiesbaden, Germany). Cells were seeded at 1 × 10^6^ cells/mL and cultured in 10 mL medium per 75 cm^2^ cell culture flask or 1 mL/well in 12-well tissue culture plates (both TPP, Trasadingen, Switzerland) at 37 °C and 5% CO_2_. Growth medium and cytokines were replaced on day 3. On day 5, maturation was induced by adding growth medium supplemented with 10 ng/mL rhIL-1β, 1000 U/mL rhIL-6, 1 mg/mL PGE_2_, and 10 ng/mL rhTNF (all obtained from R&D Systems, Wiesbaden, Germany). On day 7, the adherent and suspension cells were harvested and subjected to flow cytometry analysis and functional assays.

### 2.3. Co-Cultures and T Cell Proliferation Assay

For treatment of myeloid cells, 5 × 10^8^ to 7.5 × 10^9^ EV particles/mL or PBS (control) were added to 2 × 10^5^ or 1 × 10^6^ CD14^+^ and CD33^+^ myeloid cells cultured in a final volume of 200 µL or 1 mL medium, respectively.

T cell proliferation was assessed by adapting previously reported standardized protocol [33]. CD14-depleted PBMCs were generated from the CD14-depleted fraction obtained during the CD14^+^ cell isolation, as described in the previous section. One day prior to co-culture, CD14-depleted PBMCs were thawed and cultured in 48-well plates overnight in RPMI 1640 medium supplemented with 5 U/mL rhIL-2 (Novartis). 1 × 10^7^ CD14-depleted PBMCs were labeled with 10 µM eFluor 450 Dye (Thermo Fisher Scientific, Waltham, MA, USA) according to the manufacturer’s instruction. CD14-depleted PBMCs were seeded at 50,000 cells/well in 96-well tissue culture plates (TPP, Trasadingen, Switzerland) and stimulated with 1 µg/mL coated anti-CD3 (clone OKT3, BioLegend, San Diego, CA, USA), 5 U/mL rhIL-2 and 2 µg/mL soluble anti-CD28 (Miltenyi Biotec, Bergisch Gladbach, Germany). moDCs were cultured with allogeneic CD14-depleted PBMCs at a ratio of 1:8. Conditioned medium and cells were collected after co-culturing for 4 days at 37 °C, 5% CO_2_. The proliferation of CD3^+^CD4^+^ and CD3^+^CD8^+^ T cells was assessed in quadruplicates by flow cytometry. Conditioned medium was analyzed by cytokine multiplex assay, as described below.

### 2.4. Cytokine Single- and Multiplex Assay

Cytokines and chemokines in conditioned medium from cell cultures were analyzed using the Bio-Plex Pro Human Chemokine TNF-α Set (171BK55MR2) and Bio-Plex Human Cytokine Screening Panel, 48-plex panel (both Bio-Rad, Munich, Germany) as described previously [34]. Conditioned medium was collected, centrifuged at 2000× *g* for 10 min at 4 °C, and stored at −80 °C until further analysis. Samples were measured in duplicates using the Bio-Plex^®^ 200 system (171000201, Bio-Rad, Munich, Germany). Cytokine concentrations were determined with the Bio-Plex^®^ Manager Version 6.2 (Bio-Rad, Munich, Germany).

### 2.5. Flow Cytometry

Phenotypes of myeloid and T cells were assessed by flow cytometry as previously reported [30,34]. Briefly, CD14^+^ and CD33^+^ moDCs were stained with anti-CD14 (REA599), CD33 (REA775), CD80 (REA661), CD83 (REA714), CD86 (REA968), HLA-DR (REA805) and respective isotype (IT) antibodies. CD14-depleted PBMCs were stained with anti-CD3 (REA613), CD4 (REA623) and CD8 (REA734) antibodies. Dead cells were excluded by propidium iodide or DAPI staining. EVs were bound to 3.9 µm beads (Invitrogen, Thermo Fisher Scientific, Waltham, MA USA) and stained with anti-CD63 (REA1055), CD81 (REA513), or respective IT as reported previously [14]. All antibodies and dyes were purchased from Miltenyi Biotec, Bergisch Gladbach, Germany. EVs and cells were acquired on FACS Calibur (BD Biosciences, Heidelberg, Germany) and MACSQuant^®^Analyzer 10 (Miltenyi Biotec, Bergisch Gladbach, Germany). Data were analyzed by FlowJo V10.7.2. (BD Biosciences, Heidelberg, Germany).

### 2.6. RNA Isolation, Whole Transcriptome Sequencing, and Bioinformatics

Total RNAs from CD14^+^ cells were isolated using the mirVana miRNA isolation kit (AM1561), followed by removal of DNA (Turbo DNA-free Kit, AM1907) and RNA quantification (Qubit RNA high sensitivity assay kit, Q32852, all Thermo Fisher Scientific, Waltham, MA, USA). RNA 6000 Pico chips (Agilent Technologies, Santa Clara, CA, USA) were used to analyze RNA integrity. Strand-specific RNA-seq libraries were constructed using the KAPA RNA HyperPrep Kit with RiboErase (HMR) (8098140702, Roche, Basel, Switzerland). The quality of libraries was confirmed using the Agilent High Sensitivity DNA assay, and sequenced to about 100M reads per sample on an Illumina NovaSeq at a minimum of 2 × 100 bp.

Paired-end strand-specific 100-bp reads were aligned to human genome build 38 (hg38/GRCh38) using the Bowtie2 aligner v.2.3.5.1. Ribosomal and tRNA reads were filtered, and secondary alignments and reads with mapping scores less than 30 were removed. Transcript expression was quantified as the number of fragments per kilobase million (FPKM) with cufflinks suite (http://cole-trapnell-lab.github.io/cufflinks/) (accessed on 15 December 2020). Results were plotted using R statistical environment (https://www.R-project.org/) (accessed on 1 December 2020). and figures were compiled using the ggplot2 package. Deconvolution of myeloid cell sub-populations from the whole transcriptome data was performed using CIBERSORT [35]. The lists of M1, M2, DC, and IFN signature genes, were obtained from the Hallmark gene sets curated by the Molecular Signature Database (MSigDB, http://www.gsea-msigdb.org/gsea/msigdb/collections.jsp) (accessed on 1 March 2020). as well as assembled based on previously reported gene sets [36,37].

### 2.7. Statistical Analysis

Statistical analysis and graphs were performed with GrapPadPrism 9 (GraphPad Software, San Diego, CA, USA). Normal distribution was analyzed with the Shapiro-Wilk test. Differences between conditions were calculated by paired or unpaired two-tailed student’s t-test or one-way ANOVA with subsequent Turkey adjustment regarding mean and standard deviation (SD). Nonparametric values were analyzed with Mann-Whitney U test. *p* values ≤ 0.05 were considered statistically significant (* *p* ≤ 0.05, ** *p* ≤ 0.01, *** *p* ≤ 0.001, **** *p* ≤ 0.0001).

## 3. Results

### 3.1. EwS EVs Induce Pro-Inflammatory Responses in Myeloid Cells

We sought to determine if EwS EVs may play a role in the observed accumulation of immunosuppressive M2 macrophages [7,8,10] and CD14^+^HLA-DR^low/neg^ monocytes [23] in the tumor microenvironment and blood circulation of EwS patients. For EV isolation, the conditioned medium from A4573, A673, TC32, and TC71 EwS cell lines was subjected to ultrafiltration and differential centrifugation. In parallel, we purified EVs from MRC5 fibroblasts and from the blood plasma of healthy donors, to be used as negative controls. As assessed by nanoparticle tracking analysis (NTA), the median size of the EV preparations from EwS and MRC5 cells and healthy donors’ plasma, ranged from 100 to 170 nm (Figure 1A). As shown by immunoblotting and flow cytometry, these EV preparations were positive for classical exosome markers CD63, CD81, and TSG101 and negative for the frequent EV contaminant calnexin (Figure 1B,C and Appendix A), consistent with characteristics of small EVs [13]. In addition, we identified other common EV constituents, including the protein chaperons Hsp90 and Hsp70, and the cytoskeletal proteins tubulin and actin (Figure 1B and Appendix A). We also noticed some variability in the protein composition between independently isolated EV preparations and between EVs from different EwS cell lines (Figure 1B and Appendix A), which could be due to the protein loss during EV isolation or the variable expression of these proteins in the respective parental cells, or both. The subsequent testing of EV activities was thus performed based on the NTA-quantified number of particles, to ensure standardized comparisons between the experiments [38]. We also examined independently isolated EV preparations, including those with higher and lower levels of CD81, CD63, or TSG101, to identify common properties of EwS EVs.

To assess the potential pro-inflammatory effects of EwS EVs on blood-circulating myeloid cells, we representatively used CD33^+^ myeloid cells and CD14^+^ monocytes isolated from peripheral blood of healthy donors. We first tested independently isolated EV preparations for their ability to induce secretion of tumor necrosis factor (TNF), one of the major mediators of pro-inflammatory response [39]. Analysis of the conditioned medium by cytokine singleplex assay showed a significant elevation of TNF released by CD14^+^ cells after the 6-h treatment with any of three different EV preparations from A4573, A673, TC32, and TC71 cells, but not with EVs from MRC5 cells or healthy donor plasma, or PBS control (Figure 1D).

We next performed multiplex cytokine profiling of the conditioned medium after the 24-h treatment of CD33^+^ and CD14^+^ cells with EVs from A4573, A673, and TC32 EwS cell lines. In addition to TNF, we identified the upregulation of the pro-inflammatory cytokines and chemokines IL-1α, IL-1β, IL-6, IL-8, CCL2/MCP1, CCL3/MIP-1α, and CCL4/MIP-1β, whose levels were induced by 10-1,000-fold compared to PBS control (Figure 2A,B). Similar effects were observed in both CD33^+^ and CD14^+^ cells, with A4573 EVs eliciting the strongest response. In contrast to healthy donor plasma EVs, A4573 EV-induced release of IL-6, IL-8, and TNF by CD33^+^ cells was strongly dose-dependent (Figure 2C). These cytokines were undetectable in pre-conditioned medium with added spike–in A4573 EVs (Figure 2C), excluding a possibility of cross-contamination from the growth medium or EVs themselves. Likewise, A673 EVs (but not PBS control or MRC5 EVs) induced persistent and dose-dependent secretion of IL-6 and IL-8, and the effect was evident 18 h after a short 6-h exposure to A673 EVs (Figure 2D). Despite some variations between preparations from different cell lines (e.g., A673 EVs were less efficient in inducing TNF release compared to EVs from TC32 and TC71 cells; compare Figure 2D,E), overall pro-inflammatory activities of EV preparations from A673, TC32, and TC71 cells were comparable, indicating that induction of pro-inflammatory responses is a common property of EwS EVs. However, because A4573 EVs exhibited unexplainably high induction of a large number of pro-inflammatory cytokines and chemokines (Figure 2A,B), they were considered an outlier and were excluded for the subsequent analysis.

To investigate whether EwS EVs may induce pro-inflammatory responses during the 5-day differentiation towards moDCs, CD33^+^ myeloid cells were differentiated with IL-4 and GM-CSF in the presence of EVs from TC32 cells or MRC5 fibroblasts. By day 3, levels of IL-6 and TNF were increased by 480- and 16-fold, respectively, in the conditioned medium from TC32 EV-treated CD33^+^ cells compared to PBS control (Figure 2F), while only IL-6 reached statistical significance (*p* ≤ 0.05, unpaired two-tailed *t*-test). Moreover, increased IL-6 and TNF secretion by TC32 EV-treated immature DCs persisted even after the medium change and was detected by day 5, in contrast to MRC5 EVs which exhibited much weaker effects (Figure 2F). Together, these results demonstrate a strong EwS EV-driven induction of pro-inflammatory responses in CD33^+^ and CD14^+^ myeloid cells.

### 3.2. EwS EVs Impair DC Differentiation and Maturation of CD33^+^ Myeloid Cells and CD14^+^ Monocytes

To examine if EwS EVs may preferentially affect myeloid cells at early stages of differentiation or later on, during their maturation towards moDCs, TC32, A673, or TC71 EVs were added to freshly isolated CD33^+^ or CD14^+^ cells during differentiation (days 0 and 3) or maturation (day 5), alongside with the appropriate cytokine cocktails (Figure 3A,E). Flow cytometry analysis by day 7 showed that CD14^+^ cells (Figure 3B, *p* ≤ 0.05, one-way ANOVA with multiple comparison Turkey test) and CD33^+^ cells (Figure 3D, *p* ≤ 0.05, paired two-tailed *t*-test) failed to upregulate co-stimulatory molecules CD80, CD86 and HLA-DR after being exposed to TC32, A673 or TC71 EVs at day 0, in contrast to MRC5 EVs which had no effect (Figure 3B,D and Appendix A). Compared to control CD14^+^ cells, downregulation of CD80, CD86, and HLA-DR was significant with all three independently isolated TC32 EV preparations (*p* ≤ 0.05, unpaired two-tailed *t*-test), with preparation #3 exhibiting the strongest effect (Figure 3C). Although there might be manifold reasons for the stronger activity of this preparation, compared with preparation #1 (the weakest one), it exhibited enrichment with Hsp90 and Hsp70, while having similar levels of TSG101 and reduced levels of CD81, tubulin, and actin (Appendix A). Adding TC32 EVs at later time points during differentiation (day 3) and maturation (day 5) did not significantly interfere with the expression of maturation markers (Figure 3E, Appendix A). Since independently isolated EV preparations including those from three distinct EwS cell lines exhibited similar activities, these results indicate that EwS EVs impair the maturation of myeloid cells, especially when they are exposed to EwS EVs at very early stages of differentiation.

### 3.3. EwS EVs Rewire Gene Expression Programs Associated with Inflammatory Responses and Maturation of Myeloid Cells

To identify gene expression programs affected by EwS EVs in myeloid cells, we performed a whole transcriptome analysis of CD14^+^ cells differentiated in the absence or presence of EVs from TC32 or TC71 EwS cells (Figure 4A). We found that both TC32 and TC71 EVs induced similar gene expression patterns, with 412 transcripts commonly detected in both EwS EV-treated compared to control CD14^+^ cells (Figure 4B and Appendix A). In contrast, only 228 and 282 were shared between control and either TC32 or TC71 EV-treated cells, respectively (Figure 4B). CIBERSORT analysis of the immune cell gene expression programs revealed activation of monocyte and macrophage-associated signatures in both EwS EV-treated cells, while “activated DC” signatures were downregulated (Figure 4C), supporting a potential link between EwS EVs and the reported prevalence of these subtypes in the TME of EwS [7,8,9]. Differential gene expression analysis of the canonical genes associated with M1 and M2 macrophages or DCs (Appendix A) further confirmed a coordinated switch of gene expression programs in both TC32 and TC71 EV-treated compared to control cells. Concordant with cytokine profiling, genes encoding pro-inflammatory cytokines and chemokines, such as *IL1B, IL6*, *CXCL8*/*IL8*, *TNF*, *CCL3*, and *CCL4)* were upregulated (Figure 4D, “M1” panel), alongside with immunosuppressive cytokines (*IL10* and *TGFB1*), and extracellular matrix-degrading enzymes *ADAM8* and *MMP14* (Figure 4D, “M2” panel). Also, in line with our flow cytometry data, in both TC32 and TC71 EV-treated cells we observed downregulation of *CD86* and *HLA-DRA* and *B1*, although CD80 and several other markers of maturation (*CD40*, *CD209/DCSIGN,* and *LAMP3)* were upregulated (Figure 4D, “DC” panel).

Consistent with previous reports describing tumor-derived EVs [27,28], immune checkpoints *CD47* and *CD274/PD-L1*, and key regulators of myeloid differentiation (e.g., *TRIB1*, *SOCS3*, *STAT3*) were upregulated in EwS EV-treated cells compared to control, with TC32 EVs generally eliciting stronger effect (Figure 4D). Moreover, TC32 and TC71 EV-treated CD14^+^ cells exhibited activation of interferon (IFN)-stimulated genes, with key drivers of the type I/III IFN antiviral response, including *DDX58/RIG-1*, *OASL*, *IFITM1-3*, *IFIH1/MDA5*, *IFI6*, *INFAR1*, *INFAR2*, and *MX1,* being strongly upregulated compared to untreated cells (Figure 4E). Further investigations are required to validate these findings. Taken together, EwS EVs profoundly change gene expression programs in CD14^+^ monocytes, consistent with activation of pro-inflammatory signaling, antiviral response, and impaired maturation of these cells.

### 3.4. CD14^+^ Monocytes Differentiated in the Presence of EwS EVs Suppress T Cell Activation

Sustained release of pro-inflammatory cytokines and type I/III IFNs by tumor and immune cells eventually lead to immunosuppression and tolerance, affecting both innate and adaptive immunity [40,41]. Given a persistent release of pro-inflammatory cytokines by EwS EV-treated myeloid cells and their impaired maturation, we next examined a potential impact of these cells on T cell activation, using IFNγ release and T cell proliferation as readouts. Specifically, CD14^+^ cells differentiated in the absence or presence of EVs from TC32 EwS cells or MRC5 fibroblasts were co-cultured with allogeneic CD14-depleted PBMCs and analyzed by cytokine profiling and T cell proliferation assays (Figure 5A). Compared to PBS control, we observed a significant increase in IL-6 and IL-10 levels and reduction of IFNγ in the conditioned medium from co-cultures with CD14^+^ cells differentiated in the presence of TC32 EVs (Figure 5B; *p* ≤ 0.05, Mann-Whitney U test). The effect was specific to TC32 EVs and was not observed with MRC5 EV-pretreated CD14^+^ cells. In contrast, TNF was unselectively induced by both TC32 EV- and MRC5 EV-pretreated CD14^+^ cells (Figure 5B), in line with unspecific TNF release from MRC5 EV-treated CD33^+^ cells (Figure 2F).

We next assessed the effect of EV-pretreated CD14^+^ cells on the proliferation of T cells, by co-culturing them with allogeneic CD14-depleted PBMCs. We found that compared to control, CD14^+^ cells differentiated in the presence of TC32 or A673 EVs (but not with MRC5 EVs) significantly reduced the proliferation of both CD4^+^ and CD8^+^ T cells (Figure 5C; *p* ≤ 0.001 and *p* ≤ 0.01, one-way ANOVA with multiple comparison Turkey test), with CD4^+^ T cells being stronger affected. In line with reduced proliferation, flow cytometry analysis of CD8^+^ T cells co-cultured with A673 EwS EV-treated CD14^+^ cells revealed a significant reduction of the activation marker CD69 compared to untreated control cells (*p* ≤ 0.05, one-way ANOVA with multiple comparison Turkey test), while CD25 expression was not affected (Appendix A). Although the detailed mechanisms remain to be studied, these results indicate that CD14^+^ cells differentiated in the presence of EwS EVs reduce activation of both CD4^+^ and CD8^+^ T cells, by inhibiting their proliferation and IFNγ release.

## 4. Discussion

In this study, we show that EwS EVs induce sustained pro-inflammatory signaling and impair myeloid cell maturation, affecting CD33^+^ myeloid cells and CD14^+^ monocytes. Consistent with their semi-mature phenotypes, these cells interfered with the activation of CD4^+^ and CD8^+^ T cells (Figure 6). We thus propose that EwS EVs may pathologically activate blood-circulating and tumor-infiltrating myeloid cells, thereby contributing to systemic inflammation and immunosuppression, and influencing both innate and adaptive immunity.

These conclusions are based on several lines of evidence. First, EwS EVs (but not those from healthy donor plasma or fibroblast cells) induce prolonged cytokine and chemokine release and reprogram myeloid cells at the transcriptomic level, activating pro-inflammatory and type I/III IFN response gene expression programs. In particular, EwS EVs induced the release of IL-6, IL-8, and TNF by myeloid cells, consistent with similar findings in other cancers [27,28,42,43]. Despite the change of growth medium, the release of IL-6 persisted during DC differentiation and in co-cultures of EwS EV-pretreated CD14^+^ cells with allogeneic CD14-depleted PBMCs. Furthermore, RNA-seq analysis and co-culture experiments also showed an increase of IL-10, an immunosuppressive cytokine whose production can be triggered in myeloid and Th2 cells by IL-6 [44]. Second, as demonstrated by RNA-seq and flow cytometry, CD14^+^ cells differentiated in the presence of EwS EVs exhibited a semi-mature immunosuppressive phenotype, including reduced expression of co-stimulatory molecules associated with DC maturation (*CD74*, CD86, and HLA-DR), and with MDSC phenotypes (*TRIB1*, *JAK2*, *STAT3*, *CXCR4*, *IDO1*, *ARG2*) [45,46,47]. By transcriptomic analysis, we also observed upregulation of immune checkpoint molecules *CD47* and *CD274/PD*-*L1*, which requires further investigation in terms of their impact on immunosuppressive activities of CD14^+^ monocytes. Similar effects were reported with EVs from hematological and other solid malignancies [27,28,43,48,49], indicative of common mechanisms operating in different types of cancer. Third, in accordance with their immunosuppressive phenotype, the CD14^+^ cells differentiated in the presence of EwS EVs diminished activation of CD4^+^ and CD8^+^ T cells, by reducing their proliferation and IFNγ release, with a stronger effect on CD4^+^ cells.

Although potential mechanisms remain to be studied, our data indicate that induction of immunosuppressive myeloid cells by EwS EVs may involve both protein and RNA components. Among proteins that we identified in EwS EVs, Hsp70 and Hsp90 have been implicated in the induction of pro-inflammatory responses by myeloid cells in various cancers [27,42,50,51,52]. However, based on our RNA-seq data, the major pathways commonly activated by Hsp90 and Hsp70, such as TLR2-MyD88 and TLR4, were reduced in the EwS EV-treated CD14^+^ cells compared to control. Instead, we observed the upregulation of type I/III IFN stimulated genes, including the major IFN type I/III receptors.

*IFNAR1* and *IFNAR2*, first-line antiviral defense proteins *IFITM1-3*, and the cytosolic pattern-recognition receptors *DDX58/RIG-1* and *IFIH1/MDA5*. Both RIG-1 and MDA5 are capable of recognizing pathogenic non-self RNAs and are essential components of the innate immunity activated in response to viral infections [53]. Their upregulation in both TC32 and TC71 EV-treated CD14^+^ monocytes may thus be indicative of the presence of pathogenic viral-like RNAs in EwS EVs. This possibility is supported by our recent findings showing that EwS EVs purified from patients’ plasma and TC32, TC71, and A4573 EwS cell lines are enriched with the endogenous retroviral and retroelement-derived RNAs, including *HERV-K*, *LINE-1*, *SINE/ALU*, *7SL/SRP*, and *HSAT2* RNAs [29], each of which is immunogenic and capable of inducing RIG-1 and MDA5 [54,55]. This phenomenon may therefore underlie the observed pro-inflammatory response and acquisition of immunosuppressive properties by CD14^+^ and CD33^+^ myeloid cells. In turn, the production of pro-inflammatory and immunosuppressive cytokines by these cells, including IL-6, IL-8, TNF, and IL-10, may further alter the differentiation of myeloid cells.

Overall, our study indicates that EVs released by EwS tumors may compromise both arms of the immune system, by pathologically activating myeloid cells, skewing their differentiation, and inducing immunosuppressive activity. As shown here, EwS EV-induced semi-mature moDCs can impair activation of T cells, similar to the immunosuppressive activity of MDSCs reported in other cancers and chronic infections [22,24,56]. EwS EVs may thus contribute to the observed scarce infiltration of EwS tumors with T cells, M1 macrophages, and mature DCs, as well as the prevalence of pro-tumorigenic immunosuppressive M2 macrophages and resting DCs in the EwS TME [8,10]. Furthermore, the reported expansion of abnormally differentiated immunosuppressive CD14^+^HLA-DR^low/neg^ myeloid cells [23] and CD34^+^HLA-DR^+^ IDO^+^ fibrocytes [57] in the blood of patients with EwS support a possibility that conversion of immature myeloid cells into immunosuppressive phenotypes may occur in patients’ blood, affecting early stages of myeloid cell differentiation. As such, our study uncovers a pivotal role of EwS- and, potentially, other tumor-derived EVs in immunosuppression and cancer-associated inflammation, warranting further investigation into the mechanisms of their dissemination and targeting myeloid cells in blood circulation and the local TME.

## Figures and Tables

**Figure 1 cells-10-02081-f001:**
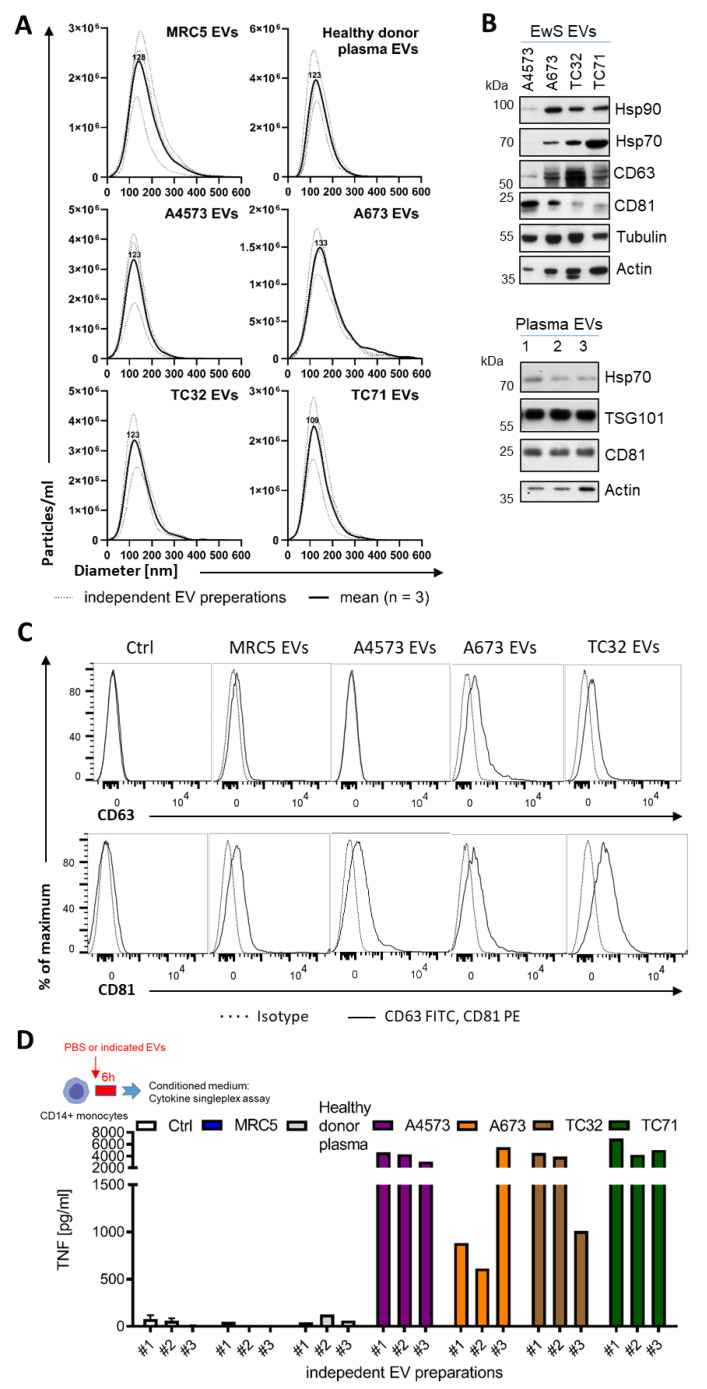
Purification and characterization of EV preparations from EwS and fibroblast cell lines and healthy donor plasma. (**A**) Size distribution of EVs isolated by ultrafiltration and differential centrifugation from healthy donor plasma, MRC5 fibroblasts, and A4573, A673, TC32, and TC71 EwS cell lines. Nanoparticle tracking analysis of three independent EV preparations (dotted line) and respective mean (solid line) is shown. (**B**) Immunoblotting-detection of chaperon proteins Hsp70 and Hsp90, EV markers CD63, CD81 and TSG101, tubulin and actin in the purified EV preparations from EwS cell lines and healthy donor plasma (*n* = 3 independent donors). (**C**) Expression of the CD63 and CD81 EV markers (solid line) compared to IT antibodies (dotted line) on MRC5 EVs or EwS EVs bound to 3.9 µm latex beads. Representative flow cytometry result (a total of 3 independent EV preparations) is shown. (**D**) TNF release from CD14^+^ monocytes treated for 6 h with three independent EV preparations (3 × 10^9^ EVs/mL) from healthy donor plasma, MRC5, or EwS cell lines. Conditioned medium was analyzed by cytokine singleplex assay.

**Figure 2 cells-10-02081-f002:**
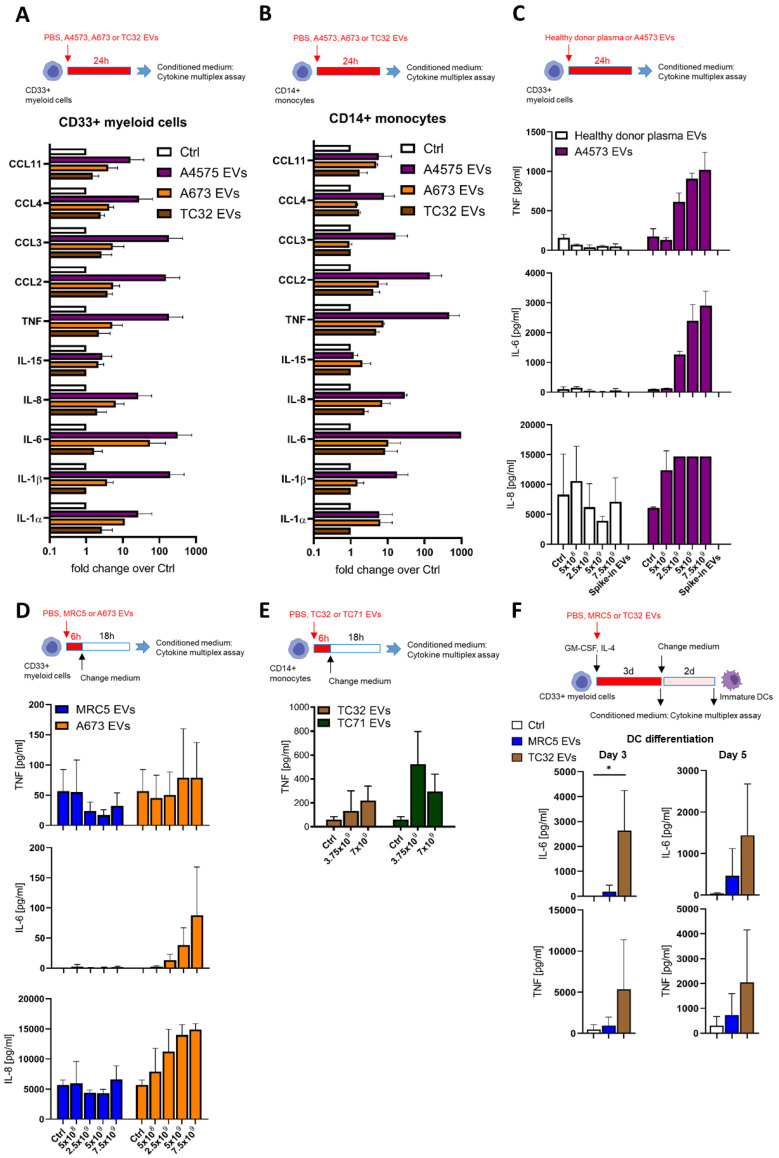
EwS EVs induce pro-inflammatory responses in myeloid cells. (**A**–**F**) Cytokine and chemokine multiplex profiling of the conditioned medium from myeloid cells treated with PBS control (Ctrl) or EVs from plasma of healthy donors, MRC5 fibroblasts, or EwS cell lines, as indicated. (**A**) CD33^+^ myeloid cells and (**B**) CD14^+^ monocytes were treated with 3 × 10^9^ EVs/mL from A4573, A673, and TC32 EwS cell lines or PBS for 24 h. (**C**) CD33^+^ myeloid cells were incubated for 24 h with 5 × 10^8^ to 7.5 × 10^9^ EVs/mL from an A4573 cell line or healthy donor plasma. (**D**) CD33^+^ myeloid cells were incubated with 5 × 10^8^ to 7.5 × 10^9^ EVs/mL from A673 EwS cells or MRC5 fibroblasts. The medium was changed after 6-h treatment and analyzed 18 h later. (**E**) CD14^+^ myeloid cells were incubated with 3.75 or 7 × 10^9^ EVs/mL from TC32 and TC71 EwS cells or PBS. The medium was changed after 6-h treatment and analyzed 18 h later. (**F**) CD33^+^ myeloid cells were differentiated with the IL-4 and GM-CSF cocktail in the presence of 3 × 10^9^ EVs/mL from TC32 or MRC5 cells, or PBS. Conditioned medium was analyzed on days 3 and 5. Results were obtained from one (**E**), two (**C**), or three (**A**,**B**,**D**,**F**) independent donors with one (**D**), two (**C**), or three (**A**,**B**,**E**,**F**) independent EV preparations. Data are presented as mean ± SD. An unpaired two-tailed *t*-test was used to calculate *p* values (**E**). * *p* ≤ 0.05.

**Figure 3 cells-10-02081-f003:**
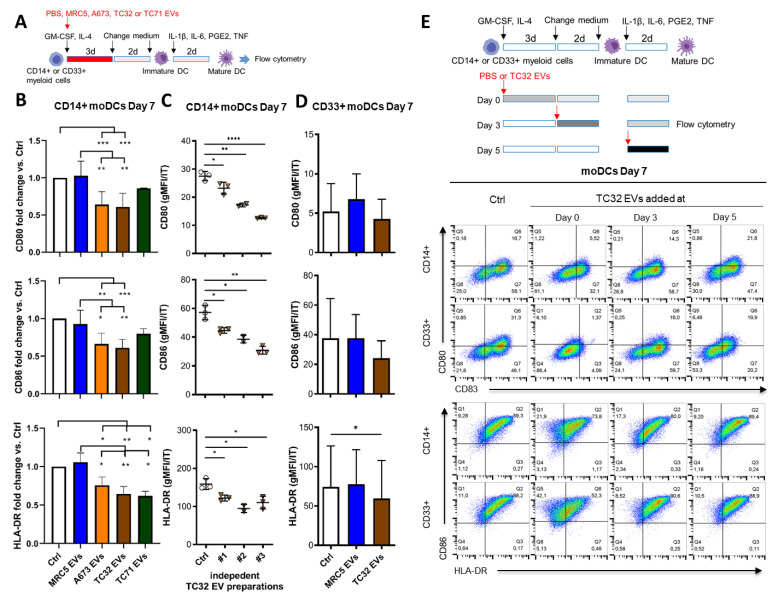
EwS EVs affect myeloid cells at an early stage of differentiation and impair their maturation. (**A**) Experimental design (**B**–**D**). CD14^+^ or CD33^+^ myeloid cells purified from healthy donors were differentiated in the presence of GM-CSF and IL-4 for 5 days and matured with the IL-1β, IL-6, PGE_2_, and TNF cocktail for additional 2 days to moDCs. The indicated EVs (3 × 10^9^/mL) or PBS control (Ctrl) were added at day 0 of differentiation. (**B**–**E**) Flow cytometry analysis of the indicated maturation markers at day 7. (**B**) Geometric mean fluorescence intensity (gMFI) of CD80, CD86, and HLA-DR normalized to IT of CD14^+^ monocytes treated with EVs from MRC5, A673, TC32, or TC71 cells, or PBS. Fold change to Ctrl is shown. (**C**) gMFI of CD80, CD86, and HLA-DR normalized to IT of CD14^+^ myeloid cells treated with three independent EV preparations from TC32 cells or PBS. (**D**) gMFI of CD80, CD86, and HLA-DR normalized to IT of CD33^+^ myeloid cells treated with MRC5 EVs, TC32 EVs, or PBS. (**E**) TC32 EVs were added to CD14^+^ or CD33^+^ myeloid cells during differentiation (day 0 and 3) or maturation (day 5). Representative pseudocolor plots of CD80, CD83, CD86, and HLA-DR compared to the respective IT antibodies are shown. The results were obtained from one (**C**) or three (**B**,**D**,**E**) independent donors with three (**B**–**E**) independent EV preparations. Data are presented as mean ± SD. One-way ANOVA with multiple comparison Turkey test (**B**), and unpaired (**C**) and paired (**D**) two-tailed *t*-tests were used to calculate *p* values. * *p* ≤ 0.05, ** *p* ≤ 0.01, *** *p* ≤ 0.001, **** *p* ≤ 0.001.

**Figure 4 cells-10-02081-f004:**
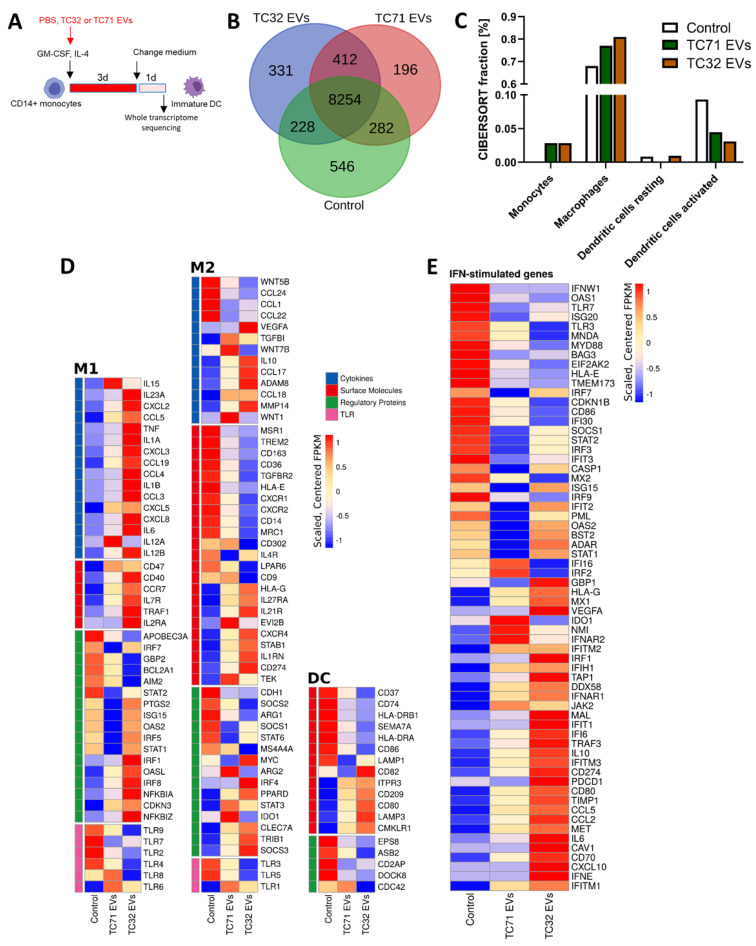
EwS EVs rewire gene expression programs associated with inflammatory responses and the maturation of myeloid cells. Experimental design (**A**) and whole transcriptome analysis (**B**–**E**). CD14^+^ monocytes were differentiated with the GM-CSF and IL-4 cocktail for 4 days in the presence of EVs from TC32 or TC71 EwS cells, or PBS (control), and subjected to RNA-seq. (**B**) Venn diagram showing an overlap between the sets of expressed genes (≥10 FPKM) in EV-treated and control cells. (**C**) CIBSESORT analysis of the identified mRNAs. The plot shows the four most represented immune cell sub-populations in EV-treated and control cells. (**D**,**E**) Heatmaps of the M1, M2, and DC (**D**) and IFN-type I/III (**E**) signature genes using the customized MSigDB Hallmark gene sets. Data represent expression values sorted according to fold change in TC32 and TC71 EV-treated CD14^+^ cells compared to control. Visualizations were done using FPKM values generated with cufflinks, scaled, and centered prior to plotting. Genes shown in (**D**) are grouped in the categories cytokines (**blue**), surface molecules (**red**), regulatory proteins (**green**), and toll-like receptors (TLR, **pink**).

**Figure 5 cells-10-02081-f005:**
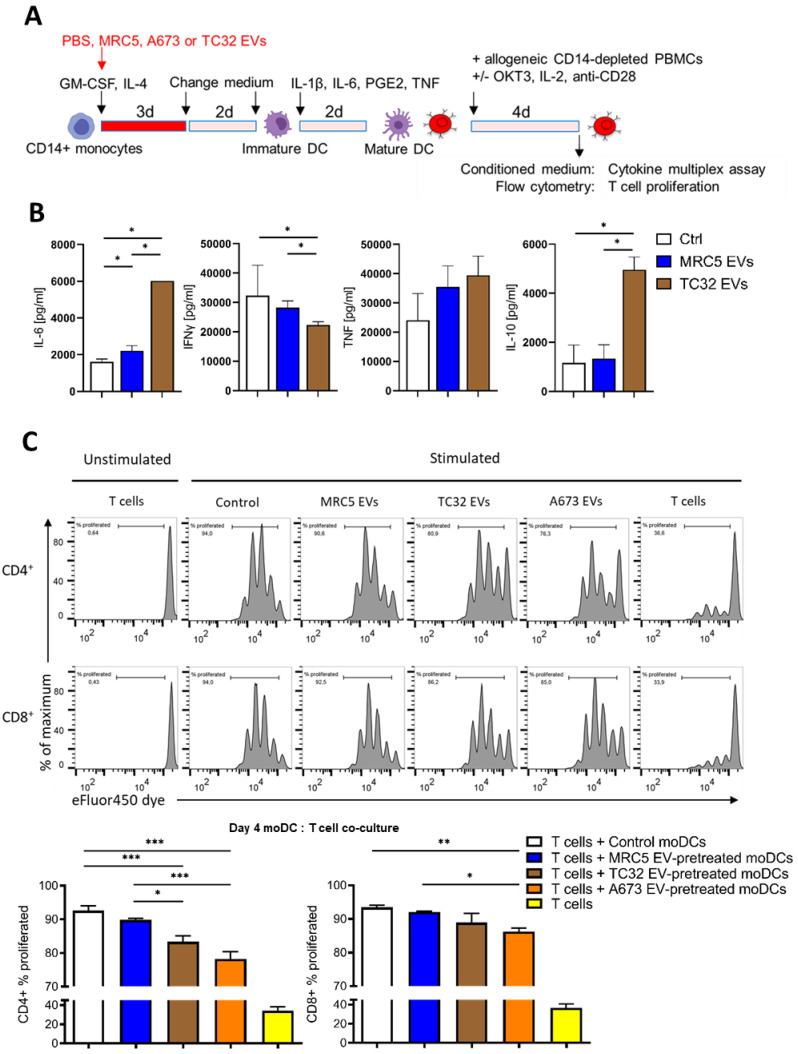
CD14^+^ monocytes differentiated in the presence of EwS EVs suppress T cell activation. Experimental design (**A**), cytokine profiling (**B)**, and flow cytometry analysis (**C**). (**B**) CD14^+^ monocytes from two donors were differentiated in the presence of 3 × 10^9^ EVs/mL from indicated cell lines and co-cultured with allogeneic CD14-depleted PBMCs for 4 days. Conditioned medium was assessed by cytokine multiplex assay. (**C**) Allogeneic CD14-depleted PBMCs were labeled with eFluor450 dye, stimulated with coated OKT3, IL2, and anti-CD28 antibodies, and co-cultured for 4 days with CD14^+^ monocytes differentiated in the presence of the indicated EVs. Representative flow cytometry results (top) and their quantification (bottom) of CD4^+^ and CD8^+^ T cell proliferation (representative of three independent donors). Results were obtained with two (**B**) or three (**C**) independent EV preparations. Data are presented as mean ± SD. Mann-Whitney U test (B) and one-way ANOVA with multiple comparison Turkey test (**C**) were used to calculate p values. * *p* ≤ 0.05, ** *p* ≤ 0.01, *** *p* ≤ 0.001.

**Figure 6 cells-10-02081-f006:**
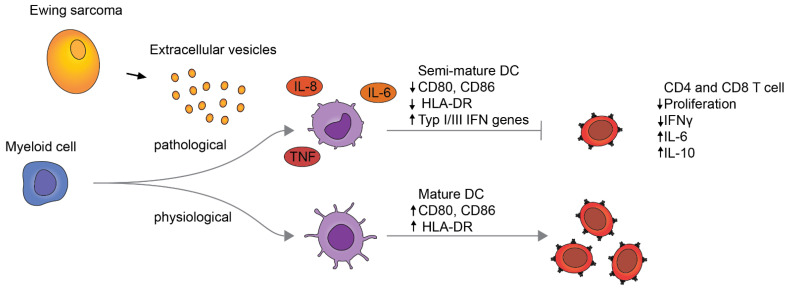
Proposed model for pathological activation of myeloid cells by EwS EVs.

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
