# Peer review of "Ewing Sarcoma-Derived Extracellular Vesicles Impair Dendritic Cell Maturation and Function"

_cells, 2021, doi:10.3390/cells10082081_

Round 1

Reviewer 1 Report

The presented study focus on extracellular vesicles and their
functional role in Ewing sarcoma, to explain the observed immunosuppressive
tumor microenvironment. For Ewing sarcoma the knowledge on these aspects
is limited and studies revealing details of the molecular
mechanisms and confirming general modi of action are helpful.

The manuscript is written in a clear and straightforward manor.
Nevertheless the presentation of the complex results could be improved
by e.g. a results table summing up the observations.

The discussion could be a bit tighter concerning the main results-
how about a graphical abstract?

If the major comments are addressed the manuscript is of interest
for the research community of Cells.

Some further comments

190  human genome build 37 (hg19/GRCh37):
for general picture this build might be valid but if details
and networks come into play this build (the basis is from 2013)
is definitely outdated. Since 2019 GRCh38 is there.
https://www.ncbi.nlm.nih.gov/grc/human
The authors should discuss this aspect in the Results.

205-206  There is a long standing statistical discussion on the usage
of SEM and SD. As can be seen by the results of the authors they
want to point to the data variablility, so they need to calculate
and show SD values.

358  'Scaled centered FPKM' as a normalization between experiments
is a weak procedure. The list of candidates will change by other
normalization methods. So Figure 3 is only giving an estimate for
that there are indeed differences. The respective text sections (Results, Discussion)
should reflect the draft nature of the candidates.

Minor issue

Supplementary Figure 1
C : Top labels partly hidden

Reviewer 2 Report

The manuscript characterizes the role of Ewing sarcoma-derived extracellular vesicles (EVs) in T cell infiltration and predominance of immunosuppressive myeloid cells. The authors successfully purified the EVs from Ewing sarcoma (EwS) cell lines, and show that the treatment of EwS EVs in myeloid cells (derived from a healthy donor) promotes the release of  pro-inflammatory cytokines , IL-6, IL-8 and TNF.  In addition, the treatment of monocyte-derived dendritic cells (moDC) with EwS EVs displayed impaired  differentiation. The authors propose that the EwS EVs inhibits CD4+ and CD8+ T cell proliferation release of and IL-10 and IL-6, and IFNg. Although the manuscript suggests the significance of EwS EVs in the activities of T cell and myeloid cells, the study has a major weakness in scientific rigor as described below.   

  • Choice of the cell lines among figures are inconsistent, thus the manuscript requires an improvement in this aspect. For example, the data obtained from TC71 should be presented in Fig1 and Fig 2.
  • Purification of the EVs is one of the most important element of this manuscript, and the presentation of the characterization requires further improvement.
  • Fig S1 should be presented as a main figure.
  • Among the multiple purified EV products, it is unclear whether the authors picked one lot (purification product) of EV in three repetitive experiments, or three independent purified EV were used in each of experiments. As shown in Fig S1B, the EVs quality are different among the different purification products. Thus, the authors should test minimum of three independent purification product per EV to test its consistency of the activity.
  • Fig S1 A: I assume the plot represents one lot of EV. All EVs used in this experiments (minimum of three purification products should be utilized) should be plotted.
  • Fig S1 B: The band intensity should be measured, normalized, and plotted with dot bot graph.
  • Which protein (or RNA) that is loaded in EVs is responsible for the activity change of T cell and myeloid cell?

Round 2

Reviewer 1 Report

The authors addressed all my concerns and improved the manuscript
remarkably.

What I do not find in the manuscript is the graphical abstract
the authors intended to include.
It should be part of the manuscript not an annex.

Reviewer 2 Report

It is a common rule to repeat the experiments for three times using independently purified EVs.  Unfortunately, this basic rule has not been followed in this particular manuscript, thus it does not meet the publication quality.  For future improvement, additional comments for the manuscript are listed below.  

Quality of experimental design requires improvement. For example, EVs from Ewing sarcoma cells and healthy plasma should be run in the same gel. This will allow us to compare the expression level difference of the proteins between Ewing sarcoma cells and healthy plasma.

The authors should present data or explain experiments clearly.  For example, the value of Y-axis of TC32 #3 in Fig 1D has been erased because of the white gap (between 1000 and 2000). The gap is also misleading, and it does not highlight the value difference between A673 #1/#2 and other sample groups. Note that the manuscript does not provide an explanation for its inconsistent value.

Fig 2D: Despite of the experiments from one-purification of EV, the error bar of the experiment is extremely large.  There may have been an technical issue when the experiment has been conducted.
